# Magnetic Field Assisted Spark Discharge-Generated Gold Nanostructures: XPS Study of Nitrogen Gas Fate and Chemical Composition of Gold Thin Films

Stefan Ručman [1,*] , Winai Thongpan [2], Wattikon Sroila [3], Niwat Jhuntama [4] and Pisith Singjai [1,4,*]

1 Department of Physics and Materials Science, Faculty of Science, Chiang Mai University, Chiang Mai 50100, Thailand
2 Department of Physics, Faculty of Science and Technology, Thammasat University, Pathumthani 12120, Thailand
3 Faculty of Science and Agricultural Technology, Rajamangala University of Technology Lanna, Chiang Mai 50300, Thailand
4 Center of Excellence in Materials Science and Technology, Chiang Mai University, Chiang Mai 50200, Thailand
* Correspondence: stefan_rucman@cmu.ac.th (S.R.); pisith.s@cmu.ac.th (P.S.)

**Abstract:** The sparking discharge process utilises high voltage to melt and evaporate tips of electrodes to create particles that can be deposited on substrate. In our research, we examine the influence of a magnetic field and nitrogen flow on gold thin-film formation onto quartz substrate. A positive effect of nitrogen flow and a 0.3 T external magnetic field was observed, in enhancement of surface plasmon band in UV visible and dispersal of nanoparticles without agglomeration. We also detected and described nitrification occurrences of gold measured by XPS at 407 eV and nitridification of quartz substrate on which gold particles are collected. These nitrogen-based chemical reactions occurred during sparking of gold wire inside of ambient air and in the magnetic field, as well during pure nitrogen flow. We measured the valence band electronic structure of gold nanoparticles deposited onto quartz substrate and found that gold thin film prepared in the magnetic field under nitrogen flow has the lowest value of 1.5 eV. Preparation of gold thin films in the magnetic field under nitrogen flow offers a highly dispersed and convenient method for productions of thin films.

**Keywords:** sparking discharge; gold nanoparticles; magnetic field; nitrification; nitrogen

## 1. Introduction

Gold nanoparticles are highly catalytic [1], have various medical applications [2], as well self-cleaning and antibacterial applications [3]; thus, it is of utter importance for society to develop a scalable and green method of nanoparticle synthesis. The sparking discharge method is a form of a physical method for synthesis of nanoparticles that does not utilise reagents, solvents or various precursors [4] and only requires gas, electrodes and electricity [5]. This gas is a barrier that an electric current needs to overcome between two electrodes, and each gas has its own threshold value of breakdown voltage at which it becomes ionised by electrical discharge. Gas temperature, pressure and composition will affect this ionisation [6]. Gas is important for particle growth because metal vapour from the high voltage of a spark discharge will collide with molecules of gas and have a cooling effect on them. However, in a spark gap, together with evaporation of electrode materials with high voltage, there is also the ionisation and breaking down of gas molecules. These molecules are in a radical state and can easily react with electrode material to form oxides, carbonites or even nitrides depending on gas composition. It was already observed that particles prepared in nitrogen are larger and can form nitrides [7]; however, usually, nitrogen should not have an effect on gold thin films prepared by sparking discharge. In this research, we investigate the difference in chemical composition of gold nanoparticles deposited onto quartz substrate by using the sparking discharge technique. We used two

types of gases in our research; one is pure nitrogen flow and the other is ambient air. Additionally, we utilised a permanent magnet to collect nanoparticles onto substrate and compared thin-film characteristics with and without a magnet, as well as in nitrogen flow and in ambient air. We already showed that using a magnetic field during the synthesis of metal thin films with a sparking discharge process can induce and enhance the crystallinity of films and, thus, increase longevity and other properties. Utilisation of a magnetic field during sparking can make the thermal annealing of substrate with a furnace redundant [8].

## 2. Experimental Setup

A custom-built spark discharge generator was used to produce gold nanoporous thin films, with and without the presence of a magnetic field, which was generated from permanent magnet strength of 0.3 Tesla, and in the presence of pure nitrogen flow or ambient air. We described our spark discharge setup previously [9] and, in Figure 1, we summarised briefly the setup in this experiment. For sparking electrodes, pure gold wires were used that were positioned 2 cm above quartz substrate that was put on a 0.3 T permanent magnet. High voltage ablates the tip of gold wire and creates nanoparticles that are deposited onto substrate. Two conditions for carrier gas were used; the first was nitrogen gas flow and the second ambient air.

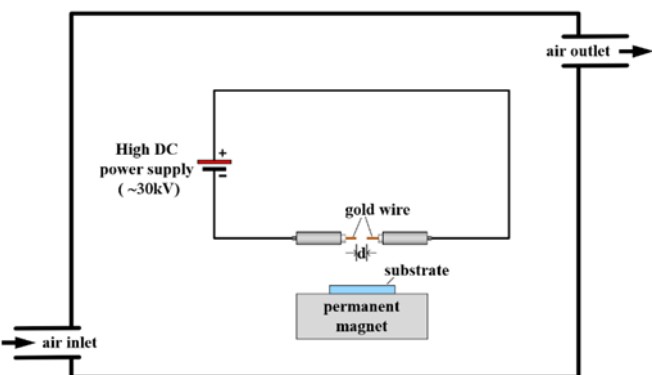

**Figure 1.** Experimental setup; high DC voltage is discharged from capacitor 24 nF.

Nitrogen composition was evaluated using XPS, Kratos Axis ULTRADLD (Kratos) spectrometer equipped with a monochromatic Al Kα X-ray source (1486.6 eV). The base pressure in the analysis chamber was approximately $5 \times 10^{-9}$ torr. The X-ray source was used with an incidence angle of 45° to the surface plane. The operation was conducted at 150 W (15 kV and 10 mA), with a spot size of $700 \times 300$ μm$^2$ and initial photo energy of 1.4 keV. The binding energy of adventitious C 1s peak at 285 eV was used for calibration of wavelength shift. The spectra were acquired (at a constant take-off angle of 90°) with the pass energy of 20 eV and analysed with the energy step of 0.1 eV using VISIONII (version 2.2.9) software. The optical absorption spectra were carried out in the range of 200 to 1100 nm using UV–vis spectrophotometer (Varian Carry 50 C). Morphological examination of nanoparticles was performed by using an FE-SEM (JEOL JSM-IT800).

## 3. Results and Discussion

Measurement of the absorption band in thin films prepared by sparking allows the study of changes in the optical properties based on the condition applied during synthesis. The absorption spectra of gold films are shown at Figure 2.

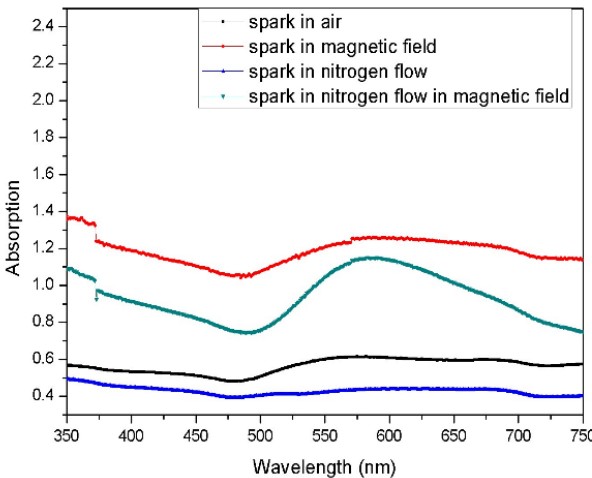

**Figure 2.** UV–vis absorption spectra of gold nanoparticles.

We can observe a longer and higher absorption value λmax to thin film prepared in nitrogen flow under a magnetic field, which might be associated with film thickness and size of gold nanoparticles. SEM characterization would tell us about the structure of thin films. Depending on aggregation of gold nanoparticles evaporated from melted wires and collected onto quartz substrate, we can notice a shift in SPR band, and the spark in the magnetic field influences aggregation of nanoparticles deposited onto quartz substrate, which influenced longitudinal peak of UV-vis absorption; thus, we can explain different plasmonic behaviour based on the preparation condition of each sample. Plasmonic peaks of samples prepared in a magnetic field with nitrogen flow and in ambient air are at 550 nm and expanded significantly compared with samples prepared outside of the magnetic field. Peak intensity increases can indicate an increase in the population of the Au atoms near the glass surfaces [10,11]. Observation of SPR peak intensity increases when nanoparticles were deposited with the magnetic field can be caused by rapid incorporation of gold nanoparticles into the quartz matrix, as observed with SEM.

Based on the results obtained from SEM (Figure 3 and Table 1), we can conclude that different morphology and size of nanoparticles are synthesised. Thin films prepared in ambient air appear flaky, as in Figure 3A, and nanoparticle size and area density vary a lot between samples, thus giving different UV-vis results. Figure 3B represents a thin film of highly agglomerated nanoparticles with clumps that are more than 100 nm and with observable primary particles of 20 nm. Interestingly, Figure 3D shows highly monodispersed nanoparticles. Here, gold was sparked in the presence of a magnetic field (0.3 T) and under flow of nitrogen air; this brings us to the conclusion that magnetic field presence and nitrogen flow have an effect on size distribution, spacing and plasmonic properties of thin films assembled.

**Table 1.** Size distribution of particles calculated from SEM image using ImageJ software.

| Name of Sample | Number of Particles | Average Size (nm) | Standard Deviation | Min and Max (nm) |
|---|---|---|---|---|
| Au wire sparked in ambient air | 53 | 65.8 | 24.8 | 25 and 130.3 |
| Au sparked in magnetic field in air | 45 | 79.7 | 24 | 33.3 and 141.5 |
| Au sparked in nitrogen flow | 50 | 37.1 | 14.8 | 12.6 and 76.7 |
| Au in nitrogen flow under magnetic field | 63 | 77.3 | 20.1 | 36.7 and 137.1 |

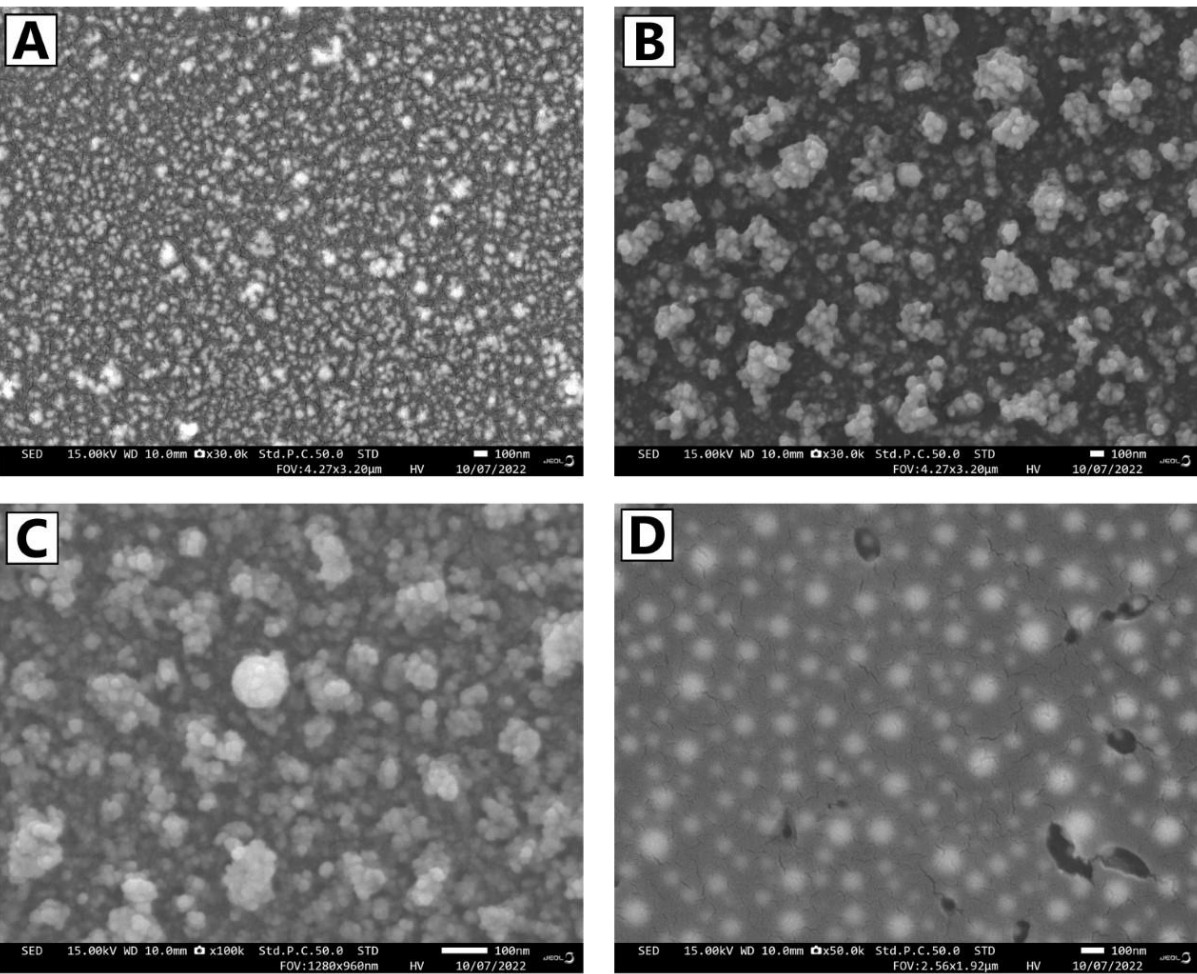

**Figure 3.** SEM image of sparked gold wire on the surface of quartz: (**A**), gold wires sparked in ambient air; (**B**), gold sparked in the magnetic field in the air; (**C**), gold wire sparked in nitrogen flow; (**D**), gold sparked in nitrogen flow under the magnetic field. Additional images available in supplementary information.

Electronic structure characterization was conducted by XPS and, as seen in previous research [12], the Au 4f spectral region can describe changes in chemical state and geometric arrangement. In our measurements, the Au 4f core level measurements from Figure 4 show three doublets with the Au 4f7/2 binding energy at 84.0 eV, 84.4 eV and 84.5 eV. Gold sparked in nitrogen and under the magnetic field exhibits binding energy, the same as bulk gold. In the rest of the conditions, the peak is shifted for 0.4/0.5 eV. Additionally, we can observe that intensity is larger with thin films prepared in nitrogen; this can maybe explain the role of nitrogen gas in agglomeration and the dispersant of gold in thin film. As described in the later section, gold, when evaporated in inert gas, changes magnetic properties, while, in oxygen (ambient air), it is diamagnetic, which may explain why concentration is lower in air.

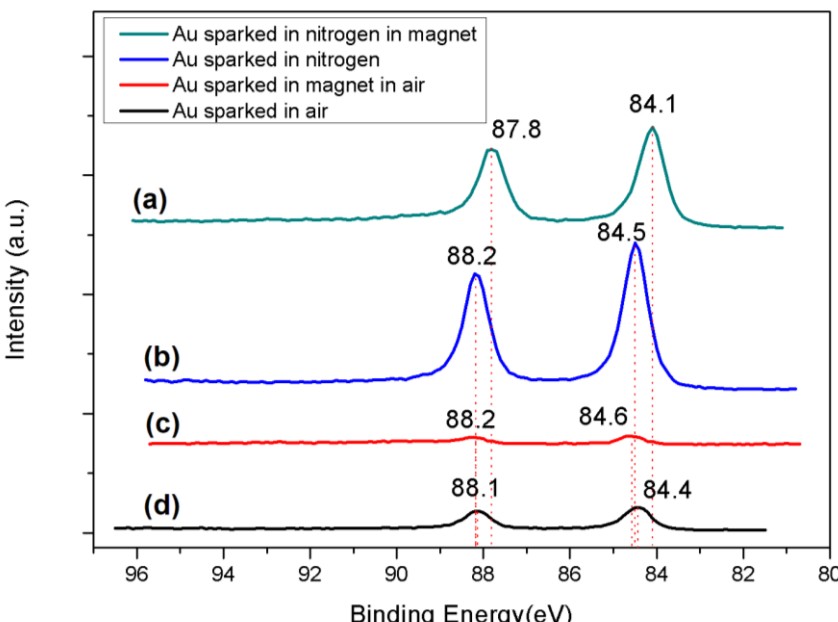

**Figure 4.** Au 4f XPS. Spectra collected under four conditions. (**a**) Gray—Au sparked in nitrogen flow under magnetic field, (**b**) blue—Au sparked in nitrogen flow without magnet, (**c**) red—Au sparked in ambient air under magnetic field, (**d**) black—Au sparked in air.

Increase in core binding energy for all samples, except when gold is sparked in nitrogen under magnetic field, can be the result of decreases in agglomeration cluster size, the result of the Coulomb attraction barrier [13] that is affected by the shape of the cluster.

Additionally, we examined binding energy of nitrogen 1s, since we noticed that nitrogen plays a role in concentration of gold in thin films. Figures 5 and 6 show the X-ray photoemission spectra of the N1s core levels for samples prepared in air inside of the magnetic field and gold wire sparked in nitrogen flow only. We found a lack of nitrogen in samples prepared in nitrogen flow with the presence of a magnetic field and samples prepared in ambient air, which maybe can be explained by a decrease in nitrogen reactivity. Samples exhibit peaks at binding energies of N1s core level spectra at 399.9 eV for a sample prepared in nitrogen flow and 400.1, 401.4, and 402.6 eV for samples prepared in air inside of a magnetic field, where these peaks are a characteristic of $N_2$, oxynitrides or carbonitrides encapsulated in thin films [14] or nitride of silicon oxide (quartz). Furthermore, the presence of 407.1 eV and 407.2 eV peak in these two samples tells, us that during sparking discharge, NOx species were created that came into reaction with gold forming nitrates.

The presence of the magnetic field enhanced the intensity of the N 1s peak, which is similar to the finding of the total valence band photoelectron spectra, represented in Figure 7. There is a large enhancement of the gold signal in the valence region of the sample prepared in the magnetic field under nitrogen flow. This is due to the difference in the amount of gold deposited onto quartz substrate.

All valence bands have somehow similar but different intensity, with the broad and flat band related to d states re-hybridized with s/p states, which extends from the Fermi level through the sharper and more intense part of the d band that spans from roughly 2 eV to 9 eV, as explained in reference [12]. Based on the extrapolation of work function value, we can observe that gold is metallic, with few variations depending on the condition in which gold was prepared. Based on theory [15], the lowest Fermi level of work function is in the sample prepared in nitrogen air under the magnetic field 1.5 eV; the valence band of gold sparked in nitrogen air and ambient air without the magnetic field are similar at 1.95 and 2.0 eV, while gold sparked in ambient air inside of the magnetic field has the highest value of 2.3 eV. These different results might be due to the effect of the magnetic field on crystallisation and formation of structure onto the surface.

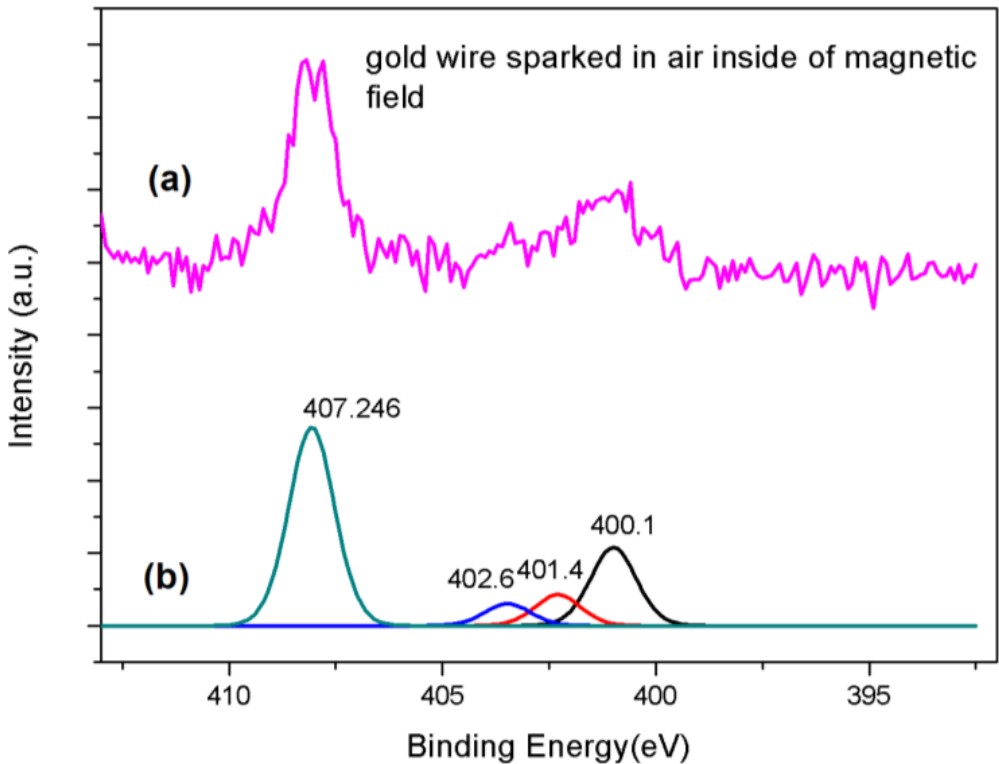

**Figure 5.** XPS N 1s spectra of thin films prepared in ambient air under magnetic field. (**a**) Pink—XPS spectra measured; (**b**) fitting of XPS.

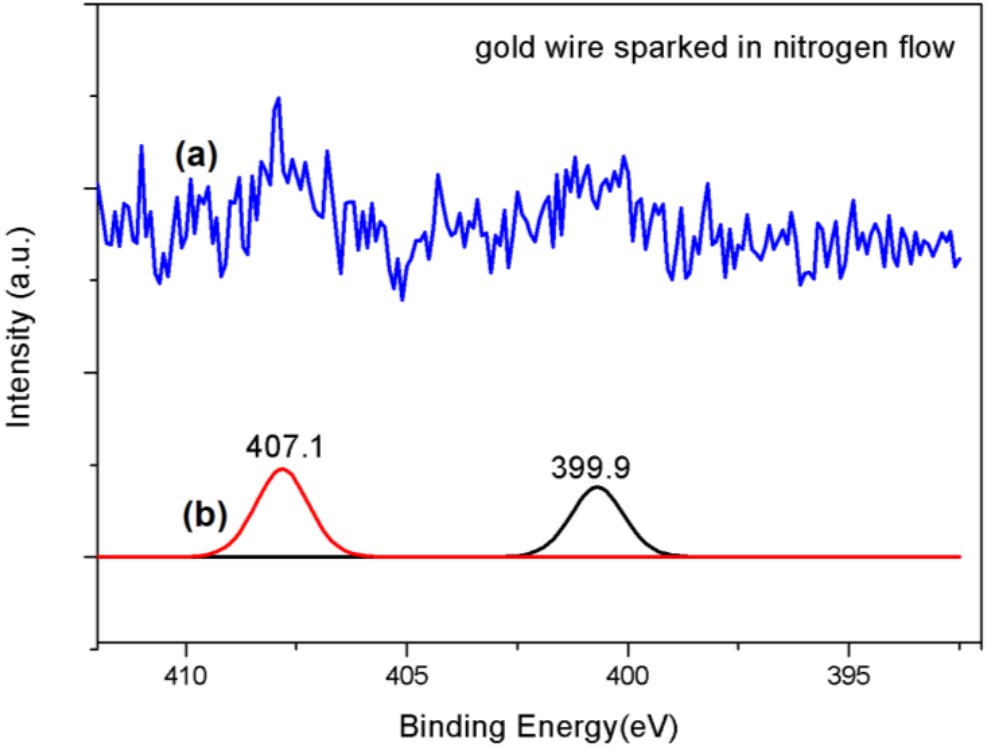

**Figure 6.** XPS N 1s spectra of thin films prepared in nitrogen flow: (**a**) blue—XPS spectra measured; (**b**) fitting of spectra.

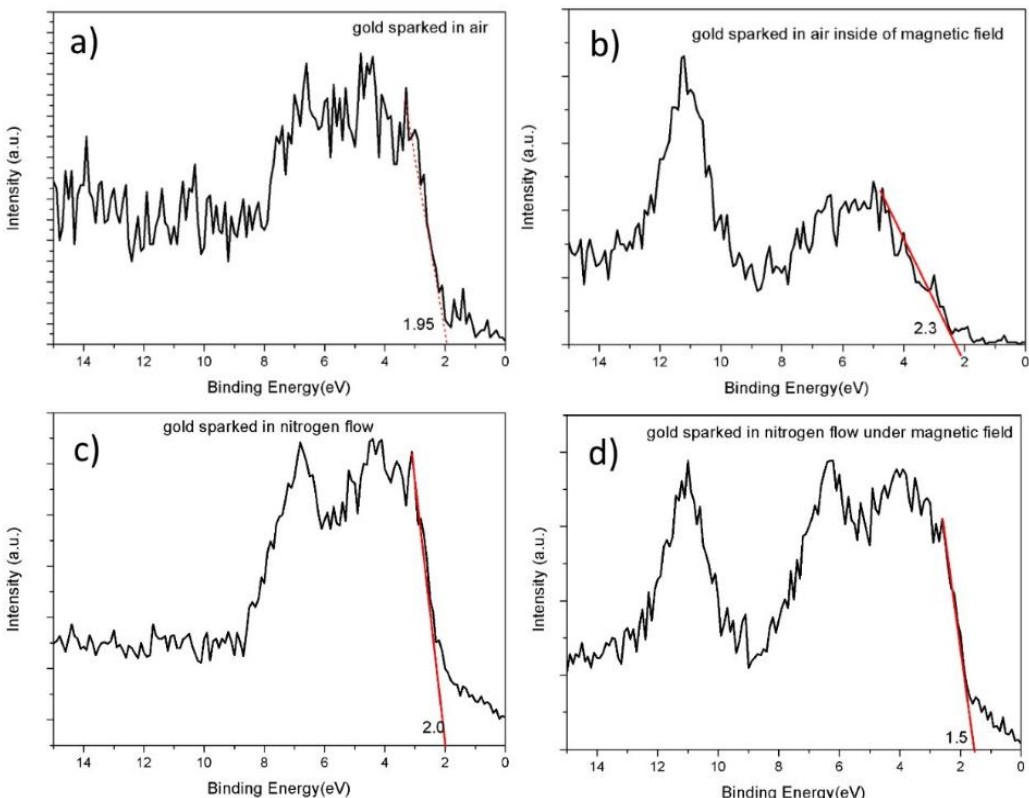

**Figure 7.** Au valence band of gold (**a**) sparked in ambient air, (**b**) sparked in ambient air inside of magnetic field, (**c**) sparked in nitrogen flow, and (**d**) sparked in nitrogen flow inside of magnetic field.

The full survey scans spectra of samples prepared in and outside of the magnetic field, as well in nitrogen flow and ambient air, and shows the preferred components, such as Cu, Zn, N, C, O, Si and Au, respectively, as shown in supplementary materials. Quantification reports of mass concentration obtained with XPS and oxidation states detected in samples are summarised in Table 2.

**Table 2.** XPS oxidation state summary of supplementary data.

| Name of Sample | O 1s | C 1s | N 1s | Au 4f | Mass Conc of Au 4f (%) |
|---|---|---|---|---|---|
| Gold sparked in ambient air | 532.288 533.593 534.846 | 284.945 285.822 286.807 287.972 289.046 | - | 84.442 88.147 | 11.14% |
| Gold sparked in magnetic field in ambient air | 531.999 532.827 533.702 | 284.900 285.907 286.808 288.015 289.078 | 400.178 401.479 402.654 407.246 | 84.570 88.245 | 4.04% |
| Gold sparked in nitrogen flow | 531.434 532.275 533.236 534.197 | 284.997 286.161 286.949 288.310 289.116 | 399.996 407.127 | 84.483 88.154 | 44.48% |
| Gold sparked in nitrogen flow with magnetic field | 531.287 532.180 533.210 534.394 | 285.010 286.228 287.052 288.127 289.058 | - | 84.091 87.829 | 28.34% |

The O-1s region can be deconvoluted into four peaks using a Lorentzian–Gaussian distribution curve. The B.E. peaks are observed at approximately 531.2 eV, 532 eV, 533.5 eV and 534.5 eV, respectively. The peak position at 531.2 eV indicates $Au^{3+}$ ($Au_2O_3$), the peak observed at 532 eV indicates $SiO_2$, the peak position at 533.5 eV indicates defects and the peak at 534.5 eV is mainly due to O deficiency [10,11]; comparing with C 1s spectra, we can observe formation of carbonates at ~289–290 eV in samples. Other C 1s peaks belong to C=O for values ~288–289 eV, and C-O bonding for values ~286.0 eV.

Effect of gas composition on noble metal nanostructure morphology and composition was already observed [16]. Gold was synthesised with sparking discharge for visible plasmonics [17]; however, substrates were annealed to up to 950 degrees Celsius to obtain broad extinction peaks. In our research findings, we obtain an ideal condition for preparation of plasmonic thin films without annealing, namely during nitrogen flow, while sparking under the magnetic field can obtain dispersed and equally shaped gold nanoparticles.

Svensson et al. [18], in their study of gold nanoparticles produced by spark discharge, discovered that nanoparticles produced are highly aggregated. Contrary to their findings, we found that aggregation can be prevented with sparking in magnetic fields under nitrogen flow, as well, we noticed the impact of air and the magnetic field on morphological characteristics of thin film.

Chemical composition of thin film consists primarily of pure gold 84 eV (in samples prepared in the magnetic field under nitrogen flow). In the rest of the samples, we observe a shift to binding energy approximately of 0.4/0.5 eV. The valence band did not vary much from 2 eV; the lowest value was recorded in the sample prepared inside of the magnetic field under nitrogen flow.

Previously, the magnetic field was used in an aerosol (gas-phase) technique to achieve directionality and control of nanoparticle assembly [19,20]. The external magnetic field employed during gas-phase synthesis was enough to compete with random Brownian forces. Chain-like aggregates were synthesised from ferromagnetic metals (Fe, Ni), and these metals in the external magnetic field have a net magnetic anisotropy in terms of an aligned dipole moment in the applied field. Since domain relaxation time of the ferromagnetic primary particle produced by a process of metal evaporation is significantly lower compared to the time period of the external magnetic field and coagulation time of gas-phase produced aerosols, this brings us to the conclusion that the domains and the resulting net dipole moment of ferromagnetic particles will be aligned with the external magnetic field during the condensation phase of primary size particles. Interaction between particles is accounted for by a pairwise interaction between magnetic dipoles aligned along the direction of the magnetic field proposed in Equation (1), as depicted in Figure 8.

$$U_{ij} = m\frac{1 - 3\cos^2\theta}{r_{ij}^3} \tag{1}$$

where $\theta$ is the angle made by the radial position vector connecting the two dipoles in the direction of the magnetic field; the parameter m has the unit of energy $k_BT_0$. The minimum condition for chain-like aggregate formation is if the repulsive part of the potential is larger than the thermal energy ($k_BT$), such that a primary particle approaching another particle along a direction perpendicular ($\theta = 90°$) to the magnetic field is successfully deterred. However, in our case we observed the formation of a single nanoparticle without agglomeration if the sample was prepared in a magnetic field under nitrogen flow, which leads us to believe that particles were repulsive no matter what direction to the magnetic field the primary particle was moving. Gold in the bulk state is diamagnetic; however, it was already observed that gold evaporated and collected in an inert atmosphere on substrate exhibits ferrimagnetic behaviour [21]; since our setup also utilised metal evaporation [22], as in Figure 8, and it was conducted under so-called inert gas, there was no nitrate formation visible in XPS, and oxygen could not react with the surface of the particle, so that can influence magnetization. It is possible that we obtained paramagnetic primary particles (in

high temperatures, for ex metal melt) that form separate nanoparticle clusters, as seen in SEM image (Figure 3D).

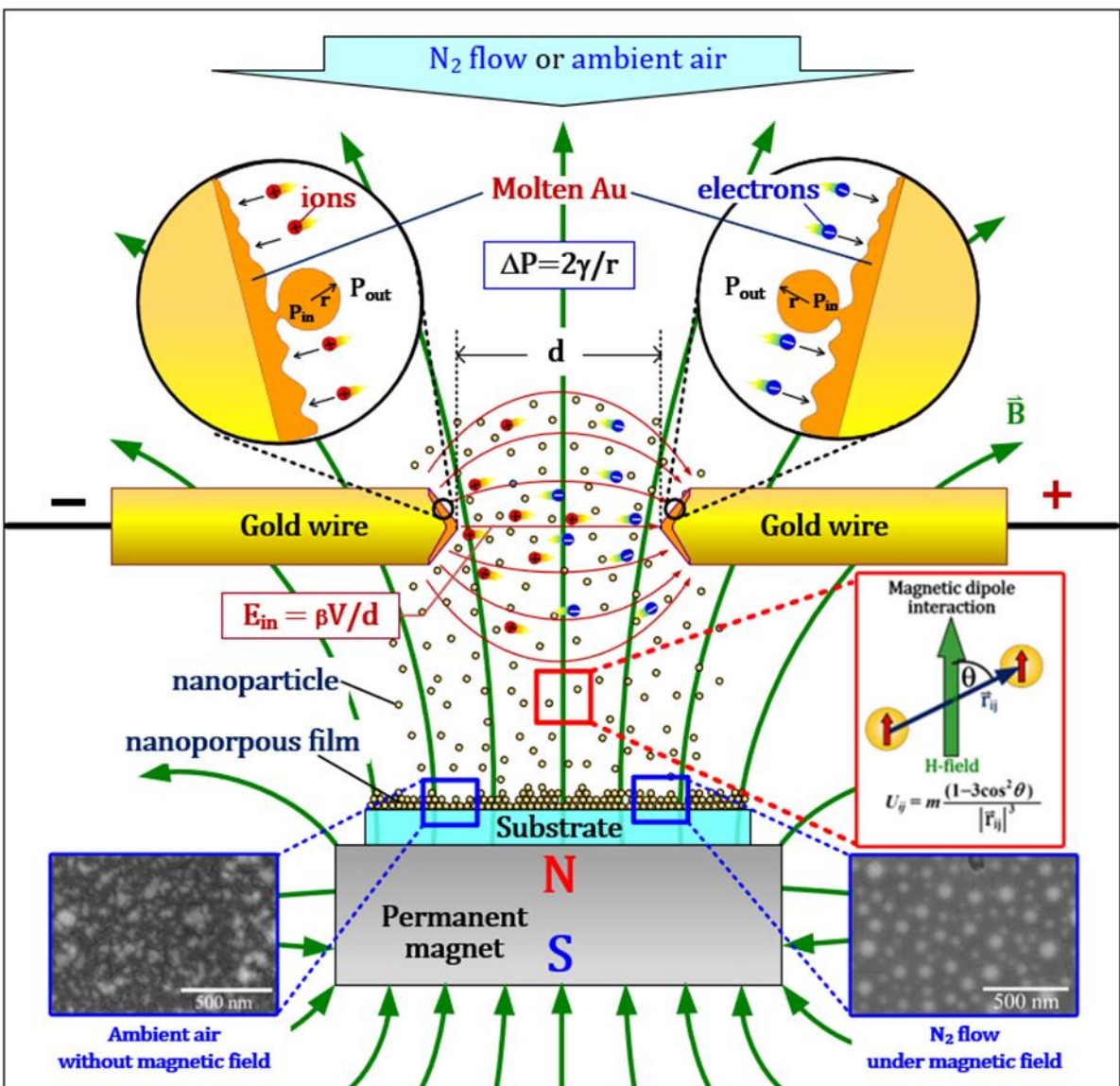

**Figure 8.** Schematic representation of chemical process in our experiment, modified from reference [20,22]. Gold wires are evaporated using high voltage and particles produced by melt were collected on quartz substrate put on permanent magnet. The high pressure and temperature on the tip surfaces were generated by the bombardment of electrons and ions. Particle size relation is described by Young–Laplace equation, i.e., $\Delta P = 2\gamma/r$, where $r$ is the radius of curvature, $\gamma$ is the surface energy of the molten metallic tip, and $\Delta P$ is the pressure difference between the inside ($P$in) and the outside ($P$out) of the particle.

## 4. Conclusions

In our research work, we prepared gold thin films over a quartz substrate by melting tips of gold wire at a high voltage using a one-step process, called spark discharge, in different conditions: with and without a magnetic field, under nitrogen flow and ambient air. The morphology and optical properties of produced thin films were investigated using SEM and UV–vis. We discovered that, when nitrogen is flown while sparking in a magnetic field, thin film with high surface plasmon effect is produced and particles are well dispersed and not agglomerated. This is seen in SEM images. We assume that observed properties of

gold are caused by different magnetic properties obtained during synthesis in pure nitrogen air. When it comes to nitrogen air reactivity, we found that nitrates were formed when gold was sparked in ambient air under magnetic field and in pure nitrogen flow without the magnetic field. Gold prepared in magnetic field under nitrogen flow also exhibits the lowest work function.

**Supplementary Materials:** The following supporting information can be downloaded at: https://www.mdpi.com/article/10.3390/magnetochemistry8120178/s1, Figure S1: SEM images of gold wire sparked in ambient air; Figure S2: XPS of gold wire sparked in ambient air; Figure S3: gold sparked in the magnetic field under ambient air; Figure S4: XPS of S3 gold sparked in the magnetic field under ambient air; Figure S5: sparked gold in nitrogen flow without magnetic field; Figure S6: XPS of sparked gold in nitrogen flow without magnetic field; Figure S7: sparked gold in nitrogen flow with magnetic field; Figure S8: XPS of sparked gold in nitrogen flow with magnetic field.

**Author Contributions:** S.R.: Writing—original draft, Conceptualization, Investigation; W.T.: Validation; W.S.: Data curation, Visualization; N.J.: Visualization; P.S.: Writing—review and editing, Supervision, Funding acquisition. All authors have read and agreed to the published version of the manuscript.

**Funding:** This research was funded by Chiang Mai University.

**Institutional Review Board Statement:** Not applicable.

**Informed Consent Statement:** Not applicable.

**Data Availability Statement:** All data is available in Supplementary Materials.

**Acknowledgments:** This research work was partially supported by Chiang Mai University.

**Conflicts of Interest:** There is no conflict to declare.

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
