# Peer review of "Magnetic Field Assisted Spark Discharge-Generated Gold Nanostructures: XPS Study of Nitrogen Gas Fate and Chemical Composition of Gold Thin Films"

_magnetochemistry, doi:10.3390/magnetochemistry8120178_

Round 1
Reviewer 1 Report
The mauscript described the preparation of gold thin films in magnetic field under nitrogen flow, and found the magnetic field played an important role on crystallization and formation of surface structure. Note the following issues should be addressed.
(1)There is a lack of mechanism analysis about magnetic field effect. In experiment, only 0.3 Tesla was employed, how about other magnetic intensities?
(2)The authors should check the figrues and English language carefully. Figure 3 and Figure 7 need reorganizing.
Author Response
Thank you for insightful comments.
reply for point 1.
We apologize for lack of mechanism analysis about magnetic field effect. We modified manuscript to add recent reference (1) Ghildiyal, P., Biswas, P., Herrera, S., Mulholland, G. W., Yang, Y., Abbaschian, R., Zachariah, M. R.; (2021). Magnetic-field directed vapor-phase assembly of low fractal dimension metal nanostructures: Experiment and Theory. The Journal of Physical Chemistry Letters, 12(16), 4085–4091. https://doi.org/10.1021/acs.jpclett.0c03463
This reference describes how magnetic field affect random Brownian forces and also describes mechanism on morphology of metal nanoparticles deposited via gas phase process. Importance of magnetic field in structure crystallization is elaborated and appropriate paragraph is added in text(conclusion section). Additionally, second possible mechanism is explained, namely, magnetron sputtering use carrier gas that was ionized just like in our experiment, it was already established in previous research (2) that flow rate plays important role in optical and morphological characteristics of thin films (3)
(2) Vijaya, G., Muralidhar Singh, M., Krupashankara, M. S., Kulkarni, R. S.; (2016). Effect of argon gas flow rate on the optical and mechanical properties of sputtered tungsten thin film coatings. IOP Conference Series: Materials Science and Engineering, 149, 012075. https://doi.org/10.1088/1757-899x/149/1/012075
(3) Ahmadipour, M., Arjmand, M., Ain, M. F., Ahmad, Z. A., & Pung, S.-Y. (2019). Effect of ar:N2 flow rate on morphology, optical and electrical properties of CCTO thin films deposited by RF Magnetron Sputtering. Ceramics International, 45(12), 15077–15081. https://doi.org/10.1016/j.ceramint.2019.04.245
regarding comment “In experiment, only 0.3 Tesla was employed, how about other magnetic intensities?”, we based this value on our previous work (4) in which we observed chemical synthesis of nitrides and this value of magnetic field was intentionally used, additionally (1) also find this value important for fractal structure formation.
(4) R., S., Jakmunee, J., Punyodom, W., Singjai, P. (2018). A novel strategy for longevity prolongation of iron-based nanoparticle thin films by Applied Magnetic Force. New Journal of Chemistry, 42(7), 4807–4810. https://doi.org/10.1039/c7nj04730d
reply for point 2.
We modified the figures to suit the publishing parameters, Figure 3 and 7 are reorginised for better quality. English language was modified following reviewer comment and abstract rewritten.
Reviewer 2 Report
This work reports preparing thin film in a nitrogen flow under magnetic fields, which is a worthy endeavor. However, the authors did not provide convincing explanation for the formation of monodispersed gold nanoparticles. The following points should be addressed before recommending the acceptance of this manuscript for publication in magnetochemistry.
1) In Figure 4, three lines (gray, blue and red) are labelled with incorrect numbers, which is not consistent with the horizontal ordinate.
2) In Figure 6, the ratio of signal-to-noise is not high enough to identify it as N 1s spectra.
3) The authors believe that nitrogen reactivity is decreased when samples were prepared in a nitrogen flow with the presence of magnetic fields and in ambient air. They need to provide more evidence and in-depth explanation for this phenomenon.
4) Page 11, Conclusions part, the temperature should be “950 degree Celsius” rather than “950 degree” .
Reviewer 3 Report
Consider a Major Revision as per my suggestions.

Round 2
Reviewer 1 Report
The manuscript has been sufficiently improved according to reviewer' suggestion, and can be published in Magnetochemistry.
Author Response
thank you for your comments
Reviewer 3 Report
The authors have improved the revised version by answering the comments positively. The quality of the present work is improved but still, a few minor suggestions are there for the authors. Kindly incorporate these suggestions as minor revisions.
Comments:
1. Can the author include a histogram graph in SEM which clears the particle distribution?
2. Figure 4 mentions the clear labels like (a), (b), etc. it is a must technical point. And, the same for figures 5 and 6.
3. Make a good presentation (quality of figure for all XPS figures). Like after figure 7 there is one more figure but no caption and labels are denoted. Take these points on a serious note.
4. Just above the conclusion author discussed the O1s fitting but there is no spectral figure mentioned kindly be serious about such fundamental mistakes.
5. The author mentioned the reference in the conclusion. I don’t think that reference should there in the conclusion. Remove it and if it is required then add it to the main text.
6. Why author presents lots of text after the conclusion better insert it in the main text in the manuscript?
Overall, have a serious look at these comments and revised it a minor revision.
Author Response
Reply.
1. Because of skewed and wide distruibution, as well high number of particles on SEM image, we use tabelar representation (Table1) instead of histogram,
Table 1 Size distribution of particles calculated from SEM image using ImageJ software
in Table 1 we sumurise number of particles, average size, standard deviation and min and max,
2. Thank you very much for comment, we just noticed this and edit by your recommendation
3. Thank you very much for the comment, Figure 7 and Figure 8 are modified to correct errors and add more information
4. We provide spectral figures in Supplementary information, however in the manuscript we summurise in Table 2. all the peaks observed by fitting. Please take a look at supplementary information
5. Thank you for the comment we revised it to follow your comment
6. We merged conclusion with discussion and write new conclusion separate from main text and without references.